# Synthesis of Novel 1-Oxo-2,3,4-trisubstituted Tetrahydroisoquinoline Derivatives, Bearing Other Heterocyclic Moieties and Comparative Preliminary Study of Anti-Coronavirus Activity of Selected Compounds

**DOI:** 10.3390/molecules28031495

**Published:** 2023-02-03

**Authors:** Meglena I. Kandinska, Nikola T. Burdzhiev, Diana V. Cheshmedzhieva, Sonia V. Ilieva, Peter P. Grozdanov, Neli Vilhelmova-Ilieva, Nadya Nikolova, Vesela V. Lozanova, Ivanka Nikolova

**Affiliations:** 1Faculty of Chemistry and Pharmacy, University of Sofia St. Kliment Ohridski, 1 J. Bourchier Avenue, 1164 Sofia, Bulgaria; 2Bulgarian Academy of Sciences, The Stephan Angeloff Institute of Microbiology, 26 Georgi Bonchev Street, 1113 Sofia, Bulgaria; 3Department of Medical Chemistry and Biochemistry, Medical Faculty, Medical University-Sofia, 2 Zdrave Street, 1431 Sofia, Bulgaria

**Keywords:** tetrahydroisoquinolines, homophthalic anhydride, imines, anti-coronavirus activity, HCoV-OC43, HCoV-229E

## Abstract

A series of novel 1-oxo-2,3,4-trisubstituted tetrahydroisoquinoline (THIQ) derivatives bearing other heterocyclic moieties in their structure were synthesized based on the reaction between homophthalic anhydride and imines. Initial studies were carried out to establish the anti-coronavirus activity of some of the newly obtained THIQ-derivatives against two strains of human coronavirus-229E and OC-43. Their antiviral activity was compared with that of their close analogues, piperidinones and thiomorpholinones, previously synthesized in our group, with aim to expand the range of the tested representative sample and to obtain valuable preliminary information about biological properties of a wider variety of compounds.

## 1. Introduction

The unceasing development and identification of new effective therapeutics for the market in recent decades has defined the tetrahydroisoquinoline (THIQ) core, widespread in nature [1], as a target structure of particularly high interest. It has been repeatedly demonstrated that the variety in substitution and the appropriate modification of the THIQ-scaffold can successfully induce a wide range of biological properties: antitumor [2,3], anticonvulsant [4], anti-HIV [5], anti-hepatitis C virus [6], antidiabetic [7], anticoagulant [8], anti-inflammatory [9], and anti-Alzheimer [10]. Floyd at al. described the absence of cytotoxicity toward mammalian cells and promising in vitro antiparasitic activity of THIQ-4-carboxamides against multiple resistant strains of P. falciparum [11,12].

The reported promising in vitro efficacy of antimalarial drugs chloroquine and hydroxychloroquine against SARS-CoV-2 in the early stage of COVID-19 pandemic [13,14] led us to the idea that such antiviral activity deserves to be looked for also in compounds that are close in their structure to antiparasitic THIQ-carboxamide (+)-SJ733 and thus provoked synthesis and investigation of the anti-coronaviral properties of novel THIQ-derivatives, containing amide or amidomethyl function in the 4-th isoquinoline core position (Figure 1). 

Our synthetic strategy for the preparation of the target 1-oxo-2,3,4-trisubstituted tetrahydroisoquinolines is based on the reaction of homophthalic anhydride and imines, allowing the isoquinoline ring closure in one step and at the same time-introduction of desired pharmacophore groups at the 2-nd and 3-rd positions and prone to further transformations carboxyl group at 4-th. The mentioned synthetic approach interaction between cyclic anhydrides and imines gives an access to a wide range of functionalized THIQ-derivatives and their structural analogues with high potential for physiological action and diversity in biological properties. That explains the continued interest to thorough studies on the chemical and stereochemical course of the reaction in each individual case [15,16]. 

Here we described the synthesis of novel THIQ-4-carboxamides and THIQ-4-amidomethyl derivatives, obtained through two different routes for the carboxyl groups transformation in the starting acids, emphasizing the chemistry and diastereoselectivity of some of the reactions used applying theoretical methods as well.

Some of the compounds synthesized were included into comparative preliminary analysis of their antiviral activity against the replication of two strains of human coronavirus-229E and OC-43. In order to shed light on the anti-coronavirus potential of a larger number of differently substituted compounds, the representative group subjected to these initial biological tests was expanded with THIQ-derivatives, piperidinones, and thiomorpholinones previously synthesized by us [17,18,19]. This initial comparative study lead us to the selection of a THIQ-based structure with the most promising anti-coronavirus properties to be modified in future investigations so that its antiviral activity can be improved.

## 2. Results and Discussion

### 2.1. Synthesis

On the bases of the results for the biological activity of the previously synthesized THIQ-derivatives and piperidinones, additional heterocyclic moieties were selected. For the incorporation of the indole heterocycle, the reaction between homophthalic anhydride (**1**) and imine **2** in boiling toluene (45 min) was investigated (Figure 1). If the reaction mixture was refluxed for longer, the mixture became darker and the yield diminished. Acid **3** crystallized from the reaction mixtures upon cooling, but, instead of its direct isolation, we preferred to dissolved it firstly in 10% NaOH as this facilitated the transformation of the *cis* diastereomer into *trans*. The crude acid is sufficiently pure for further transformations and was used without additional purification.

Initially, an attempt was made to synthesize the planned carboxamides of the acid *trans*-**3**, by employing SOCl_2_ in chloroform, as was performed in the past [18]. The reaction mixture quickly darkens and after the evaporation of the solvent, and addition of the corresponding amine did not give the desired product. The conversion of acid *trans*-**3** into the target carboxamides was then tried with *N*,*N*’-diisopropylcarbodiimide (DIC) as a coupling agent. In the case of thiomorpholine and pyrrolidine, the only product was the result of O to N migration of the O-acyl moiety of the activated ester species. Although the quantity of such byproduct should be suppressed by the nucleophilicity of the amine and the use of DCM as a solvent [20], in our reactions this was the major product. The reaction was significantly prolonged in the presence of *N*-methylpiperazine as well, and *trans*-**4a** was the major product, but in this case *trans*-**4e** was formed and isolated in 15% yield. The use of lower temperature should have suppressed the formation of *trans*-**4a** [20], but after stirring the reaction mixture for three hours at 0 °C and additional 24 h at room temperature, the result was exactly the same. The reaction of acid **3** and imidazole produced after 1 h at room temperature the desired amide *trans*-**4d**. Due to its lower solubility in DCM the amide **4d** was isolated by filtration, but the resulting solid contained significant amounts of 1,3-diisopropylurea. To circumvent this problem, the synthesis of amides **4d** and **4e** was repeated using 2-(1*H*-benzotriazole-1-yl)-1,1,3,3-tetramethylaminium tetrafluoroborate (TBTU) as a coupling agent. In both cases, stirring of the mixture for 1h of at 0 °C allowed us to isolate the corresponding amides in good to moderate yields.

When piperidine was used as nucleophile in the amide synthesis involving the use of DIC, the reaction was so slow that we added additional quantities of the amine after 2 days in order to facilitate the reaction. This resulted in NH-deprotonation of the indole and its subsequent nucleophilic attack on dichloromethane used as a solvent allowing the displacement of one of the chlorine atoms. This unexpected substitution was followed by nucleophilic attack of the piperidine which displaced the second chlorine atom and lead to the formation of amide *trans*-**4b**. Such tendency of DCM to react with nucleophiles upon prolonged standing at room temperature is already reported in the literature [21].

The interaction between homophthalic anhydride (**1**) and imine **5** was investigated earlier in our group, and it was found that the reaction is highly stereoselective toward *trans*-isomer of the expected acid when it is carried out in boiling 1,2-dichloroethane (DCE) as a solvent, and the main product is (±)-*trans*-**6** (75% yield) [22].

In order to obtain the *cis*-isomer of the acid-**6** in sufficiently high yields to ensure its further stereoselective conversions into *cis*-derivatives, the biological activity of which is also of interest to be investigated and compared with that of their *trans*-diastereomers, the reaction between anhydride **1** and imine **5** (Figure 2) was carried out at room temperature, using the same solvent as a reaction media. After the end and workup of the reaction mixture, two acidic products were isolated and separated by column chromatography.

The ^1^H NMR spectrum of one of them corresponded to that obtained for the *trans* isomer of acid **6** [22]. Analysis of the ^1^H NMR spectrum of the second acidic compound showed the presence of two different pairs of signals for the characteristic protons H-3 and H-4 of the acid-**6**-two doublets at 5.33 ppm and 4.41 ppm with spin–spin coupling constant ^3^*J* = 1.0 Hz, and another two doublets with integral intensities lower than 0.5 appearing at 5.19 ppm and 4.73 ppm, respectively with, ^3^*J* = 6.1 Hz. Since the relative configurations of isoquinolonic acids may be assigned on the basis of their ^1^H NMR spectra, the observed doublet signals with coupling constant for the second acidic compound (^3^*J* = 6.1 Hz) was attributed to *cis*-isomer of **6** in agreement with the reported data about *cis*-isoquinolonic acids [23,24]. Our results confirmed the isolation of both diastereomers of acid-**6**, but were also evidence of the rapid epimerization in solution of the *cis*-isomer into *trans*. Until recently, the literature data showed that the conversion of the thermodynamically more unstable *cis*-isomer of this type of compound into *trans* is possible, but during prolonged refluxing in acetic acid or shorter boiling in dimethylformamide [24]. Recently, Burdzhiev and co-workers reported that the *trans*/*cis* ratio changes upon standing of the mixture of diastereomeric isoquinolonic acids for several days at room temperature in favor of the thermodynamically more stable *trans*-isomer [17]. Such epimerization towards the *trans* diastereomer is favored when the sample is heated or when the diastereomeric mixture of acids is isolated after alkali workup with sodium hydroxide. 

The inability to isolate *cis*-isomer of the acid **6**, due to the ongoing rapid epimerization in solution, in contrast to previously investigated cases [16,19] where *cis*-isomer was isolated (the substituents at 3-rd isoquinoline ring position were *N*-methyl-1*H*-pyrrol-2-yl or furan-2-yl), led us to the suggestion that such change takes place with the participation of the pyridine heterocycle.

To explain the observed spontaneous conversion of *cis*-**6** to *trans*-**6**, we performed a systematic theoretical study comparing the stability of diastereomeric acids **6** using the Gaussian 09 software package [25]. The stability of *cis*-**6** acid and different conformers of *trans*-**6** (Figure 2) were examined. The relative energy differences between the studied conformers relative to the most stable one are summed in Table 1. Each of the studied structures is optimized by hybrid meta-GGA DFT functional M06-2X [26] in combination with 6-31 + G(d,p) [27] basis set in gas phase and dichloroethane solution. Frequency calculations for each optimized structure were performed at the same level of theory M06-2X/6-31 + G(d,p). No imaginary frequencies were found for any of the optimized structures.

The thermodynamic computations showed that the most stable form is *trans*-**6a**, in which the substituents at C8 (C-4) and C9 (C-3) are antiperiplanar. *Trans*-**6a** conformer is the most stable in gas-phase and solvent dichloroethane. The optimized geometries for *cis*-**6** and *trans*-**6a** conformers are shown in Figure 3. *Trans*-**6a** is more stable than *cis*-**6** by 4.1 kcal/mol. A hydrogen bond is formed in *cis*-**6** conformer between the acidic proton and the lone pair of N atom in pyridine (Figure 3). The energy difference going from gas phase to dichloroethane solution becomes smaller. The reason is the higher dipole moment of *cis*-**6,** and it is expected that this energy will be even smaller in media with higher polarity.

The optimized geometry of *trans*-**6a** is close to the crystal structure. The calculated dihedral angle C15-C8-C9-C10 is 162.5° in solution and 163.4° taken from the X-ray diffraction analysis of a crystal [22]. The dihedral angle H8-C8-C9-H9 between the vicinal protons H8 (H-4) and H9 (H-3) is −70.1°/72.13° and corresponds to the conformation with their synclinal arrangement. Based on the data obtained from the ^1^H NMR spectra, X-ray structural analysis, and theoretical calculations, it can be considered that the preferred conformation of *trans*-**6** is the same in solution and in the crystal structure, namely *trans*-**6a**.

In an attempt to explain the rapid epimerization observed in this compound, and to verify our assumption that this occurs with the participation of the piperidine substituent, we modeled a possible transition state in which the pyridine nitrogen atom withdraws the H8 proton from *cis*-**6** (Figure 4) and subsequently, after rotation around the C8-C9 bond, donates proton and leads to the formation of the *trans*-diastereomer.

The calculated activation free energy difference, **ΔG≠,** for the epimerization from *cis*-**6** diastereomer is 48.3 kcal/mol and it is a possible explanation for the observed fast epimerization.

The thermodynamically more stable *trans*-isomer of acid **6** was converted into corresponding methyl ester *trans*-**7** (Figure 3) following the procedure reported earlier in the literature [28]. Through selective reduction of the ester functional group in *trans*-**7** with lithium borohydride at room temperature and tetrahydrofuran (THF) as a solvent, the corresponding alcohol *trans*-**8** was obtained. Such reaction conditions do not affect the lactam carbonyl function and, at the same time, do not change the configuration of C-3 and C-4 stereogenic centers.

The reduction of ester *trans*-**7** under ultrasonic irradiation confirmed the significant shortening of the reaction time from 20 to 1.5 h, an effective modification previously reported by Burdzhiev and co-authors [17], without this affecting the yield of the target alcohol. Using the hydroxymethyl compound *trans*-**8** as an alkylating reagent in a Mitsunobu reaction, the corresponding phthalimidomethyl derivative *trans*-**9** was successfully obtained. The conversion of phthalimidomethyl substituent in 4th position of *trans*-**9** into primary aminomethyl group in *trans*-**10** was achieved under mild conditions in a presence of ethylenediamine [29]. 

In the last modification step, the acylation of the primary amino group in *trans*-**10** (Figure 4) was carried out with acyl chloride of the *N*-trifluoroacetyl-l-phenylalanine (Tfa-Phe-OH) or using *N*,*N*’-dicyclohexylcarbodiimide (DCC) as a carboxylic group activating agent in the cases of *N*-protected l-proline (Boc-Pro-OH) and l-methionine (Tfa-Meth-OH). Our attempt to apply acyl chloride method to the previously mentioned amino acids resulted in complex mixtures of products which were difficult to separate for further purification. Compounds **11**–**13** were isolated and characterized as mixtures of diastereomeric products (**a** and **b**).

Compounds **3**, **4**, **8**–**10**, **11**–**13a** + **b** are new, and their structure and *trans* relative configuration were established on the bases of 1D and 2D NMR spectral data (Appendix A).

### 2.2. Virology

Compounds **4a**-**e** and **11a** + **b,** together with already known 1-oxo-2,3,4-trisubstituted THIQ-derivatives **Avir**-**1**–**4**,**6**,**7** piperidinone, **Avir**-**8,** and thiomorpholinone **Avir**-**5,** were included into preliminary comparative analysis of their antiviral activity against the replication of two strains of human coronavirus: 229E and OC-43. The selection of differently substituted in 2-nd, 3-th, and 4-th positions tetrahydroisoquinolinones and other heterocyclic compounds (Table 2) was driven by the need for initial information about the antiviral potential of a more diverse group of tested compounds.

Chloroquine and hydroxychloroquine were used as reference substances. For more accurate assessment of the antiviral activity, and to avoid the toxic effects of substances on cells, the cytotoxic effect of the selected compounds on the MRC-5 and HCT-8 cell lines was determined in advance (Table 3).

Compared to the MRC-5 cell line, all tested substances showed significantly lower toxicity—from four to twelve times lower compared to that of chloroquine and hydroxychloroquine. The lowest toxicity was showed by **Avir**-**6** (CC_50_ = 729 µM), which was more than twelve times weaker than the reference substances. Weak toxicity was demonstrated by **4a** and **Avir**-**5**, as well as **Avir**-**3** (about 10 times weaker than chloroquine and hydroxychloroquine), followed by **Avir**-**2** (CC_50_ = 550 µM) and **Avir**-**8** (CC_50_ = 515 µM).

The lowest toxicity regarding the HCT-8 cell line was shown by **4e** (CC_50_ = 724 µM) and **4c** (CC_50_ = 645 µM), followed by **Avir**-**3** (CC_50_ = 588 µM), **4d** (CC_50_ = 580 µM), **Avir**-**6** (CC_50_ = 579 µM) and **Avir**-**5** (CC_50_ = 522 µM).

Comparing the cytotoxicity of the tested heterocyclic compounds showed that toxicity on the MRC-5 cell line was generally lower.

After determining the range of non-toxic concentrations of the test substances, their anticoronaviral activity against both studied strains was monitored.

Five heterocyclic compounds showed activity against coronavirus strain 229E replication; the strongest inhibition was demonstrated by **Avir-7** (SI = 560), which was almost similar to that of chloroquine (SI = 600). **Avir-8** anticoronaviral activity was also noticeable (SI = 367), but it was almost twice as weak as chloroquine. **Avir-4** (SI = 23), **11a** + **11b** and **Avir-5** (SI = 22) have almost the same activity relative to each other (about 30 times lower than that of chloroquine). The substances **4a**, **4c**, **4d**, and **Avir-1**, as well as hydroxychloroquine, did not show any activity against the replication of coronavirus strain 229E (Table 3).

Considering the influence of the studied heterocyclic compounds on the replication of coronavirus strain OC-43, the highest activity was shown by **Avir-8** (SI = 972.0), which demonstrated the strongest activity compared to all tested substances (including reference ones), against replication and of the two viral strains included in the study. **Avir-7** (SI = 280.0) showed almost three and a half times lower antiviral activity compared to **Avir-8**. In addition, **4c** (SI = 35.8), **Avir-6** (SI = 15.2), and **Avir-5** (SI = 7) also had effect on the replication of the OC-43 strain. The remaining heterocyclic compounds studied did not affect the replication of that virus strain (Table 3).

If we compare the antiviral activity of the tested substances against the replication of the two strains of coronavirus, we can mark that a larger number of substances affect the replication of 229E to varying degrees. However, as mentioned earlier, the strongest inhibition was demonstrated by **Avir-8** against the observed replication of OC-43. In both strains studied, two of the substances were observed to be the most active: **Avir-7** and **Avir-8**, with **Avir-7** showing stronger inhibition in strain 229E and **Avir-8** in strain OC-43.

After determining the effect of the tested substances on the replication of coronavirus strains 229E and OC-43, we focused on the possibility that the tested heterocyclic compounds have a protective effect on still healthy cells (cell lines MRC-5 and HCT-8) preceding viral infection and on the adsorption phase of coronavirus tested strains.

For this purpose, the substances with the best pronounced antiviral properties were applied in their MTC on uninfected cells from MRC-5 or HCT-8 cell lines for different time intervals (15, 30, 60, 90, and 120 min). In the first studied time interval, **Avir-1** showed significant protection with a decrease in viral titer by Δlg = 2.0. A slight effect with Δlg = 1.5 was also shown by **Avir-2** and **Avir-3**. With increasing time of exposure, the activity of **Avir-1** was preserved with a pronounced effect of Δlg = 2.0 to 120 min. The activity of **Avir-2**, **Avir-3**, and **Avir-6** remained weak for up to 120 min (Δlg = 1.5) (Table 4).

It was found that a larger number of substances have a protective effect on cells in the treatment of healthy HCT-8 cells with the studied heterocyclic compounds and subsequent infection with strain OC-43. At 15 min of exposure, **Avir**-**1**, **Avir**-**2**, and **Avir**-**3** showed significant activity (Δlg = 2.0). **Avir**-**1** and **Avir**-**2** retain their activity (Δlg = 2.0) for up to 120 min. **Avir**-**3** decreased its activity during the next studied time interval, and at 120 min its protection was within the range of viral titers Δlg = 1.0. Some of the compounds, such as **11a + 11b, Avir**-**5, Avir**-**7**, and chloroquine, do not have a protective effect on cells (Table 5).

We found that some of the studied heterocyclic compounds during a certain incubation period managed to cause a protective effect on still healthy cells. This leads to a reduction in the amount of virus that enters the cell, with subsequent infection, which reduces the viral yield as a whole. We also wanted to see if this effect persists or changes in any way if both the test compounds and the virus particles (which are trying to attach to the host cell) affect the cells at the same time.

It was found that within 15 min of exposure, none of the test substances had an effect and, in general, viral adsorption was not impaired when monitoring the adsorption step of virus strain 229E on MRC-5 cells. The effect was very weak at 30 min of exposure. Significant effect was observed with **Avir**-**1**, **Avir**-**6**, and **Avir**-**8** (Δlg = 1.75) only after exposure of 60 min (Table 6). With increase in the exposure time of the heterocyclic compounds to 120 min, in some of them the effect intensifies, this effect being most pronounced in **Avir**-**6** (Δlg = 2.5) (comparable to that of chloroquine). The influence on the adsorption stage of **Avir**-**1**, **Avir**-**2**, **11a + 11b**, **Avir**-**5**, **Avir**-**7**, and **Avir**-**8** (Δlg = 2.0) was also significant. **Avir**-**3** and **Avir**-**4** have a weak effect on the adsorption step of coronavirus strain 229E (Table 6).

When considering the effect of the studied derivatives on the adsorption of coronavirus strain OC-43 on sensitive HCT-8 cells, the same dependence was observed as for strain 229E and MRC-5 cell line. At 15 and 30 min, the effect is absent or weak. Significant effect was observed at 60 min of exposure, the most noticeable being in **Avir**-**1**, **Avir**-**2**, and **Avir**-**6** (Δlg = 2.0), as well as in **Avir**-**7** and **Avir**-**8** (Δlg = 1.75). Again, with increasing exposure time, the effect increases with some of the compounds. At 120 min, the most pronounced effect, equal to that of chloroquine, was shown by **Avir**-**1**, **11a + 11b**, and **Avir**-**7** (Δlg = 2.5). With similar activity were **Avir**-**6** and **Avir**-**8** (Δlg = 2.34), and **Avir**-**2** and **Avir**-**5** (Δlg = 2.25). **Avir**-**3** and **Avir**-**4** (Δlg = 1.5) showed low activity (Table 7).

The experiments conducted in a system free of virus showed that some of the studied heterocyclic compounds have a protective effect on the membranes of HCT-8 and MRC-5 cells. The observed effect is most likely the result of binding or structural modification of the test substances with cell membrane surface structures, which are essential for viral entry. The effect does not depend much on the time of exposure, although, with some of the studied substances, it increases slightly over time. It is possible that such an effect is due to the inhibitory effect of the test substances on the adsorption stage of viral replication. In this case, the activity of the substances was monitored in the presence of coronavirus virions. The results we obtained generally show that the effect on the adsorption step is slightly more pronounced compared to the protective effect on healthy cells, and is manifested in a larger number of the studied heterocyclic compounds. This also implies some influence on viral structures necessary for the virus to attach to the sensitive cell, which enhances the reported inhibitory effect.

2D-QSAR Hansch analysis [31] was performed using the Codessa 3.3.1 software. The relationship between the structure of the studied compounds and their antiviral activity against the 229E strain was quantitatively examined. The molecular geometry of tetrahydroisoquinoline derivatives (structures **Avir 1–4**, **6**, **7**, and **11**) in the series was optimized using the DFT approach with M062X functional [26] in combination with the 6-31G(d,p) basis set. All DFT calculations were performed with the Gaussian16 program package [25]. The biological activity is presented as the negative decimal logarithm of the data in Table 3. The multiple linear regression (MLR) method was used to find the optimal correlation between antiviral activity and computed physicochemical descriptors. About 300 theoretical descriptors were calculated for each molecule, classified into several groups: (i) constitutional, (ii) topological, (iii) geometric, (iv) quantum chemical, (v) thermodynamic, and (v) electrostatic.

The best equation obtained is given below: (1)$$\log\left(\frac{1}{IC_{50}}\right)=0.838\;\pm \times FMZNC+2.757\left(\pm 0.308\right),$$
n=6,  =0.901,  F=36.5,  =0.077,  =0.828

The independent variable FMZNC stands for fractional minimum Zefirov negative charge, which is the maximum negative atomic charge times its atomic surface area divided by the total negative charge. The high values of the coefficient of determination R² (0.901), the Fisher F-value (36.5), and the cross-validated coefficient of determination q² (0.828) are indicative for the predictive power of the obtained correlation.

The mechanism of antiviral activity of tetrahydroisoquinoline derivatives is not yet clear, but QSAR analysis allows us, based on the descriptors involved in the equation, to derive a working hypothesis for the factors influencing biological activity of the molecules. FMZNC is an electronic descriptor reflecting the charge distribution in molecules. It can be concluded that the electrostatic interactions between the respective molecule and the target biomolecule play an important role in the mechanism of biological activity.

## 3. Materials and Methods

### 3.1. General Information

All solvents used in the present work are HPLC grade and commercially available. The starting materials are commercially available and they were used as supplied. Melting points of the compounds with crystal structure were determined on a Boetius PHMK 0.5 apparatus and are uncorrected. NMR spectra (^1^H-, ^13^C-NMR) were obtained on Bruker Avance DRX-250 (250.13 MHz), Bruker Avance NEO 400 (400.23 MHz), Bruker Avance III HD (500.13 MHz), and Bruker Avance II 600 (600.13 MHz) spectrometers. The chemical shifts are given in ppm (δ) using tetramethylsilane (TMS) as an internal standard or the residual solvent peak [32]. Some of the compounds are isolated in the form of diastereomeric mixtures. In these cases, the signals for the equivalent atoms from the different diastereomers in the NMR spectra are denoted with “a” for the signals at lower frequency and “b” for the signals at higher frequency.

The elemental analyses of the compounds were carried out at the Laboratory of Elemental Analyses at the Faculty of Chemistry and Pharmacy (University of Sofia) and the Microanalytical Laboratory of IOCCP (BAS). Liquid chromatography mass spectrometry analysis (LC-MS) was carried out on a Q Ex-active® hybrid quadrupole-Orbitrap® mass spectrometer (ThermoScientific Co, Wal-tham, USA) equipped with a HESI® (heated electrospray ionization) module, Tur-boFlow® Ultra High Performance Liquid Chromatography (UHPLC) system (Thermo-Scientific Co, Waltham, USA) and HTC PAL® autosampler (CTC Analytics, Zwingen, Switzerland). 

Human colon carcinoma (HCT-8) cells were purchased from the American Type Culture Collection (ATCC). Permanent HCT-8 [HRT-18] (ATCC-CCL-244, LGC Standars) were maintained at 37 °C and 5% CO_2_ using sterile RPMI 1640 (Roswell Park Memorial Institute Medium, ATCC-30-2001) supplemented with 0.3 g/L l-glutamine (Sigma-Aldrich, Darmstadt, Germany), 10% horse serum (ATCC-30-2021), 100 UI penicillin, and 0.1 mg streptomycin/mL (both Sigma-Aldrich, Darmstadt, Germany).

Diploid cell line MCR-5, derived from normal lung tissue, was purchased from the American Type Culture Collection (ATCC). Cells were grown at 37 °C and 5% CO_2_ using Eagle’s Minimum Essential Medium (Lonza), supplemented with 10% fetal bovine serum and (Gibco) 100 IU penicillin and 0.1 mg streptomycin / mL (Sigma-Aldrich). The cells were incubated at 37 °C, in the presence of 5% CO_2_.

Human Coronavirus OC-43 (HCoV-OC43) (ATCC: VR-1558) strain was propagated in HCT-8 cells in RPMI 1640 supplemented with 2% horse serum, 100 U/mL penicillin, and 100 μg/mL streptomycin. Cells were lysed 5 days after infection by double freeze and thaw cycles, and the virus was titrated according to the Reed and Muench formula. Virus and mock aliquots were stored at −80 °C.

Human coronavirus 229E (ATCC: VR-740) strain was replicated in monolayer MRC-5 cells in Eagle’s Minimum Essential Medium supplemented with 2% fetal bovine serum, 100 U/mL penicillin, and 100 μg/mL streptomycin. The cells were incubated with the virus for 5 days and then lysed by double freezing and thawing. The virus was titrated by the method of Reed and Muench. Viral aliquots were stored at −80 °C.

### 3.2. Synthesis

#### 3.2.1. Preparation of rel-(3R,4R)-3-(1H-indol-3-yl)-2-(2-methoxyethyl)-1-oxo-1,2,3,4-tetrahydroisoquinoline-4-carboxylic acid (**3**)

To a solution of 1-(1*H*-indol-3-yl)-*N*-(2-methoxyethyl)methanimine (**2**, 202.26 g/mol, 23 mmol) in dried toluene (40 mL), homophthalic anhydride (**1**, 3.729 g, 162.14 g/mol, 23 mmol) was added. The reaction mixture was refluxed for 45 min until the bottom layer became transparent. Acid **3** crystallized from the reaction mixtures upon cooling and could be obtained after filtration. The solid precipitate was dissolved in 10% NaOH (30 mL) and the water layer was washed with ethyl acetate until the organic phase stopped yellowing. The alkaline layer was acidified with 15% HCl, and the resulting acid solidified but could not be filtered. The water layer was decanted and the solid was suspended in ethyl acetate (100 mL). The crystalline phase was filtered and dried to yield 5.331 g (64%) of acid **3** as whitish powder. The water layer was extracted with ethyl acetate (3 × 15mL), and the organic phase was combined with the filtrate and was dried (Na_2_SO_4_). The solvent was evaporated under reduced pressure and the residue was triturated with acetonitrile (10 mL) and filtered to give additional 1.252 g of acid **3**, thus giving overall yield of 79%. m.p. (ethyl acetate) 206–207 °C. 

^1^H NMR (DMSO-d_6_, 500,13 MHz) δ 3.09 (dt, 1H, NC*H*_2_, *J* = 6.4, 13.8 Hz), 3.21 (s, 3H, OC*H*_3_), 3.45 – 3.55 (m, 2H, C*H*_2_O), 4.15 (dt, 1H, NC*H*_2,_
*J* = 5.9, 13.8 Hz), 4.19 (d, 1H, H-4, *J* = 1.5 Hz), 5.68 (br. s, 1H, H-3), 6.64 (d, 1H, C*H*N-Ind, *J* = 2.3 Hz), 7.03 (ddd, 1H, C*H*-Ind, *J* = 1.0, 6.8, 8.2 Hz), 7.09 (ddd, 1H, C*H*-Ind, *J* = 1.0, 6.8, 8.2 Hz), 7.16–7.22 (m, 1H, H-5), 7.32 (d, 1H, C*H*-Ind, J = 8.1 Hz), 7.36–7.41 (m, 2H, H-6, H-7), 7.51 (d, 1H, C*H*-Ind, J = 7.9Hz), 7.93–7.98 (m, 1H, H-8), 10.87 (d, 1H, N*H*, J = 2.3 Hz), 12.97 (br. s, 1H, COO*H*); ^13^C NMR (DMSO-d_6_, 125.76 MHz) δ 44.90 (1C, N*C*H_2_), 49.57 (1C, C-4), 55.81 (1C, C-3), 58.04 (1C, O*C*H_3_), 69.83 (1C, *C*H_2_O), 111.86 (1C, *C*H-Ind), 113.11 (1C, *C*-Ind), 117.93 (1C, *C*H-Ind), 119.01 (1C, *C*H-Ind), 121.46 (1C, *C*H-Ind), 122.63 (1C, *C*HN-Ind), 124.98 (1C, *C*-Ind), 126.93 (1C, C-8), 127.63 (1C, C-6), 129.00 (1C, C-8a), 129.51 (1C, C-5), 131.62 (1C, C-7), 134.62 (1C, C-4a), 136.46 (1C, *C*-Ind), 162.77 (1C, C-1), 172.20 (1C, *C*OOH). ESI-HRMS (m/z) calculated for [M + H]^+^ ion species C_21_H_21_N_2_O_4_: 365.1501; found 365.1648.

#### 3.2.2. Synthesis of Amides with DIC as Coupling Agent (General Procedure)

To a suspension of the acid **3** (0.364 g, 1 mmol) in DCM (5 mL), DIC (0.2 mL, 1 mmol) was added at room temperature. An equimolar amount of the corresponding amine was added, and the reaction mixture was stirred at room temperature until the completion of the reaction (TLC). The reaction mixture was diluted with ethyl acetate (25 mL) and subsequently washed with 1:4 HCl (3 mL), H_2_O (5 mL), 10% Na_2_CO_3_ (3 mL), and H_2_O (2 × 5 mL). The organic layer was dried over dry Na_2_SO_4_, and the solvents were evaporated under reduced pressure. The residual oil was recrystallized.

##### *rel*-(3*R*,4*R*)-3-(1*H*-indol-3-yl)-*N*-isopropyl-*N*-(isopropylcarbamoyl)-2-(2-methoxyethyl)-1-oxo-1,2,3,4-tetrahydroisoquinoline-4-carboxamide (**4a**)

In this case, the amine that was used was thiomorpholine. After two days, the resulting oil was triturated with ethyl acetate 3 mL and the oil was dissolved with heating. After cooling, crystals were formed and filtered. Yield 0.340 g (69%) white crystals, m.p. 185–187 °C. 

^1^H NMR (CDCl_3_, 500,13 MHz) δ 1.11 (d, 3H, NHCH(C*H*_3_)_2_, *J* = 6.6 Hz), 1.12 (d, 3H, NHCH(C*H*_3_)_2_, *J* = 6.6 Hz), 1.37 (d, 3H, NCH(C*H*_3_)_2_, *J* = 6.9 Hz), 1.42 (d, 3H, NCH(C*H*_3_)_2_, *J* = 6.8 Hz), 3.11 (ddd, 1H, NC*H*_2_, *J* = 4.2, 8.7, 14.3 Hz), 3.27 (s, 3H, OC*H*_3_), 3.52 (ddd, 1H, C*H*_2_O, *J* = 4.2, 4.2, 9.8 Hz), 3.66 (ddd, 1H, C*H*_2_O, *J* = 4.2, 8.7, 9.8 Hz), 3.94 (oct, 1H, NHC*H*(CH_3_)_2_, *J* = 6.8 Hz), 4.29 (ddd, 1H, NC*H*_2_, *J* = 4.2, 4.2, 14.3 Hz), 4.48–4.58 (m, 2H, H-4, NC*H*(CH_3_)_2_), 5.56 (d, 1H, H-3, *J* = 3.8 Hz), 6.70 (d, 1H, C*H*N-Ind, *J* = 2.5 Hz), 6.75 (br. s. 1H, N*H*CH(CH_3_)_2_), 6.98–7.01 (m, 1H, H-5), 7.15 (ddd, 1H, C*H*-Ind, *J* = 1.0, 7.0, 7.9 Hz) 7.20 (ddd, 1H, C*H*-Ind, *J* = 1.0, 7.0, 8.1 Hz), 7.32–7.40 (m, 3H, C*H*-Ind, H-6, H-7), 7.65 (d, 1H, C*H*-Ind, *J* = 7.9 Hz), 8.16–8.19 (m, 1H, H-8), 8.46 (br. s, 1H, N*H*); ^13^C NMR (CDCl_3_, 125.76 MHz) δ 20.61 (1C, NCH(*C*H_3_)_2_), 21.43 (1C, NCH(*C*H_3_)_2_), 22.38 (1C, NHCH(*C*H_3_)_2_), 22.40 (1C, NHCH(*C*H_3_)_2_), 43.14 (1C, NH*C*H(CH_3_)_2_), 45.18 (1C, *C*H_2_N), 48.27 (1C, N*C*H(CH_3_)_2_), 49.86 (1C, C-4), 57.02 (1C, C-3), 58.73 (1C, O*C*H_3_), 71.54 (1C, *C*H_2_O), 111.71 (1C, *C*H-Ind), 114.00 (1C, *C*-Ind), 118.47 (1C, *C*H-Ind), 120.20 (1C, *C*H-Ind), 122.46 (1C, *C*H-Ind), 123.52 (1C, *C*HN-Ind), 125.40 (1C, *C*-Ind); 127.59 (1C, C-5), 128.00 (1C, C-8), 128.11 (1C, C-6), 130.46 (1C, C-8a), 131.82 (1C, C-7), 134.85 (1C, C-4a), 136.31 (1C, *C*-Ind), 153.94 (1C, N*C*ONH), 164.31 (1C, C-1), 171.44 (1C, *C*ON). ESI-HRMS (m/z) calculated for [M + H]^+^ ion species C_28_H_35_N_4_O_4_: 491.2625; found 491.2712.

##### *rel*-(3*R*,4*R*)-*N*-isopropyl-*N*-(isopropylcarbamoyl)-2-(2-methoxyethyl)-1-oxo-3-(1-(piperidin-1-ylmethyl)-1*H*-Indol-3-yl)-1,2,3,4-tetrahydroisoquinoline-4-carboxamide (**4b**)

In this case, the amine used was piperidine. After three days of stirring, an additional 0.1 mL of the amine was added. After one day, 0.2 mL DIC and 0.2 mL amine were added. After three days, the reaction was completed. The mixture was filtered and the solution was washed as described in the general procedure. The oil was triturated with ethyl acetate 5 mL, and, after cooling, crystals were formed and collected. Yield 0.362 g (62%) slightly yellow crystals, m.p. 154–156 °C

^1^H NMR (CDCl_3_, 500,13 MHz) δ 1.13 (d, 3H, NHCH(C*H*_3_)_2_, *J* = 6.7 Hz), 1.14 (d, 3H, NHCH(C*H*_3_)_2_, *J* = 6.7 Hz), 1.20–1.26 (m, 2H, C*H*_2_), 1.39–1.45 (m, 7H, 2C*H*_2_, NCH(C*H*_3_)_2_), 1.42 (d, 3H, NCH(C*H*_3_)_2_, *J* = 6.7 Hz), 2.10–2.25 (m, 4H, CH_2_NCH_2_), 3.11 (ddd, 1H, NC*H*_2_, *J* = 4.3, 8.7, 14.5 Hz), 3.29 (s, 3H, OC*H*_3_), 3.55 (ddd, 1H, C*H*_2_O, *J* = 4.1, 4.3, 9.8 Hz), 3.68 (ddd, 1H, C*H*_2_O, *J* = 4.1, 8.7, 9.8 Hz), 3.98 (oct, 1H, NHC*H*(CH_3_)_2_, *J* = 6.7 Hz), 4.32 (ddd, 1H, NC*H*_2_, *J* = 4.1, 4.1, 14.5 Hz), 4.50 (d, 1H, H-4, *J* = 3.2 Hz), 4.52–4.62 (m, 2H, NC*H*_2_N, NC*H*(CH_3_)_2_), 4.71 (d, 1H, NC*H*_2_N, *J* = 13.3 Hz), 5.56 (d, 1H, H-3, *J* = 3.2 Hz), 6.60 (s, 1H, C*H*N-Ind), 6.80–7.10 (m. 2H, H-5,N*H*CH(CH_3_)_2_), 7.15 (ddd, 1H, C*H*-Ind, *J* = 1.0, 7.1, 7.7 Hz) 7.21 (ddd, 1H, C*H*-Ind, *J* = 1.0, 7.1, 8.3 Hz), 7.33 (ddd, 1H, H-6, *J* = 1.4, 7.5, 7.5 Hz), 7.36–7.41 (m, 2H, H-7, C*H*-Ind), 7.63 (d, 1H, C*H*-Ind, *J* = 7.8 Hz), 8.19 (dd, 1H, H-8, *J* = 1.3, 7.7 Hz); ^13^C NMR (CDCl_3_, 125.76 MHz) δ 20.75 (1C, NCH(*C*H_3_)_2_), 21.42 (1C, NCH(*C*H_3_)_2_), 22.47 (1C, NHCH(*C*H_3_)_2_), 23.52 (1C, NHCH(*C*H_3_)_2_), 23.68 (1C, *C*H_2_), 25.68 (2C, 2*C*H_2_), 43.08 (1C, NH*C*H(CH_3_)_2_), 45.39 (1C, *C*H_2_N), 48.49 (1C, N*C*H(CH_3_)_2_), 49.85 (1C, C-4), 51.45 (2C, *C*H_2_N*C*H_2_), 57.03 (1C, C-3), 58.74 (1C, O*C*H_3_), 68.38 (1C, N*C*H_2_N), 71.65 (1C, *C*H_2_O), 110.74 (1C, *C*H-Ind), 112.40 (1C, *C*-Ind), 118.40 (1C, *C*H-Ind), 119.92 (1C, *C*H-Ind), 122.09 (1C, *C*H-Ind), 125.70 (1C, *C*-Ind); 127.58 (1C, C-5), 127.89 (1C, C-8), 128.01 (1C, *C*HN-Ind), 128.08 (1C, C-6), 130.77 (1C, C-8a), 131.59 (1C, C-7), 134.80 (1C, C-4a), 137.49 (1C, *C*-Ind), 153.98 (1C, N*C*ONH), 164.37 (1C, C-1), 171.38 (1C, *C*ON). ESI-HRMS (m/z) calculated for [M + H]^+^ ion species C_34_H_46_N_5_O_4_: 588.3544; found 588.3545.

##### *rel*-(3*R*,4*R*)-3-(1*H*-indol-3-yl)-2-(2-methoxyethyl)-4-(morpholine-4-carbonyl)-3,4-dihydroisoquinolin-1(2*H*)-one (**4c**)

In this case, the amine that was used was morpholine, and the reaction was completed within 30 min. The resulting oil was triturated with ethyl acetate 5 mL, and the oil was dissolved with heating. After cooling, white crystals were formed and filtered. Yield 0.220 g (51%) white crystals, m.p. 232–233 °C.

^1^H NMR (CDCl_3_, 500, 13 MHz) δ 2.67–3.00 (m, 1H, C*H*_2_-morph), 3.15 (ddd, 1H, NC*H*_2_, *J* = 5.1, 7.8, 14.1 Hz), 3.25 (s, 3H, OC*H*_3_), 3.28–3.52 (m, 6H, 1xC*H*_2_O, 5xC*H*_2_-morph), 3.53–3.77 (m, 3H, 1xC*H*_2_O, 2xC*H*_2_-morph), 4.28 (ddd, 1H, NC*H*_2_, *J* = 4.8, 4.8, 14.1 Hz), 4.76 (d, 1H, H-4, *J* = 7.2 Hz), 5.52 (d, 1H, H-3, *J* = 7.2 Hz), 6.92 (d, 1H, C*H*N-Ind, *J* = 2.3 Hz), 6.93–6.96 (m, 1H, H-5), 7.13 (ddd, 1H, C*H*-Ind, *J* = 0.7, 7.2, 8.0 Hz) 7.22 (ddd, 1H, C*H*-Ind, *J* = 0.7, 7.2, 8.2 Hz), 7.37–7.44 (m, 3H, C*H*-Ind, H-6, H-7), 7.60 (d, 1H, C*H*-Ind, *J* = 8.0 Hz), 8.20–8.25 (m, 1H, H-8), 8.57 (br. s, 1H, N*H*); ^13^C NMR (CDCl_3_, 125.76 MHz) δ 42.54 (1C, *C*H_2_-morph), 44.20 (1C, N*C*H_2_), 46.17 (1C, C-4), 46.75 (1C, *C*H_2_-morph), 57.17 (1C, C-3), 58.88 (1C, O*C*H_3_), 66.27 (1C, *C*H_2_-morph), 66.88 (1C, *C*H_2_-morph), 71.02 (1C, *C*H_2_O), 112.08 (1C, *C*H-Ind), 113.14 (1C, *C*-Ind), 118.39 (1C, *C*H-Ind), 120.34 (1C, *C*H-Ind), 122.63 (1C, *C*H-Ind), 124.33 (1C, *C*HN-Ind), 125.38 (1C, *C*-Ind); 126.43 (1C, C-5), 127.90 (1C, C-6), 128.56 (1C, C-8), 129.86 (1C, C-8a), 132.04 (1C, C-7), 135.56 (1C, C-4a), 136.54 (1C, *C*-Ind), 164.41 (1C, C-1), 169.60 (1C, *C*ON). ESI-HRMS (m/z) calculated for [M + H]^+^ ion species C_25_H_28_N_3_O_4_: 434.2080; found 434.2170.

#### 3.2.3. Synthesis of Amides with TBTU as Coupling Agent (General Procedure)

To a suspension of the acid **3** (0.182 g, 0.5 mmol) in DCM (5 mL), TBTU (0.160 g, 0.5 mmol) was added at room temperature. The corresponding amine was added, and the reaction mixture was stirred at room temperature until the completion of the reaction (TLC). The reaction mixture was diluted with DCM (25 mL) and subsequently washed with NaCl brine (3 × 3 mL). The organic layer was dried over dry Na_2_SO_4_, and the solvent was evaporated under reduced pressure. 

##### *rel*-(3*R*,4*R*)-4-(1*H*-imidazole-1-carbonyl)-3-(1*H*-indol-3-yl)-2-(2-methoxyethyl)-3,4-dihydroisoquinolin-1(2*H*)-one (**4d**)

In this case, imidazole (0.034 g, 0.5 mmol) was used as an amine. At the end of the reaction (2 h), part of the product precipitated and was filtered. The filtrate was worked as described in the general procedure, and the residual oil was triturated with ethyl acetate 2 mL; the crystals formed were collected by filtration. Yield 0.083 g (41%) of white crystals, m.p. 162–164 °C.

^1^H NMR (CDCl_3_, 400, 23 MHz) δ 2.99 (ddd, 1H, NC*H*_2_, *J* = 3.2, 10.0, 14.3 Hz), 3.04 (s, 3H, OC*H*_3_), 3.46 (ddd, 1H, C*H*_2_O, *J* = 3.2, 3.2, 9.7 Hz), 3.60 (ddd, 1H, C*H*_2_O, *J* = 2.9, 9.7, 10.0 Hz), 4.33 (ddd, 1H, NC*H*_2_, *J* = 2.9, 3.2, 14.3 Hz), 4.71 (d, 1H, H-4, *J* = 2.0 Hz), 5.81 (d, 1H, H-3, *J* = 2.0 Hz), 6.63 (d, 1H, C*H*N-Ind, *J* = 2.6 Hz), 7.01 (d, 1H, H-5, *J* = 7.5 Hz), 7.18–7.25 (m, 3H, 2xC*H*-Ind, 1x*H*-Imi), 7.35–7.39 (m, 1H, C*H*-Ind), 7.42 (ddd, 1H, H-6, *J* = 1.5, 7.5, 7.5 Hz), 7.49 (ddd, 1H, H-7, *J* = 1.2, 7.5, 7.7 Hz), 7.58–7.62 (m, 1H, C*H*-Ind), 7.66 (dd, 1H, *H*-Imi, *J* = 1.5, 1.5 Hz) 8.26 (dd, 1H, H-8, *J* = 1.5, 7.7 Hz), 8.39 (s, 1H, H-Imi), 8.75 (br. s, 1H, N*H*); ^13^C NMR (CDCl_3_, 100.64 MHz) δ 45.52 (1C, N*C*H_2_), 50.39 (1C, C-4), 56.50 (1C, C-3), 58.91 (1C, O*C*H_3_), 72.36 (1C, *C*H_2_O), 112.11 (1C, *C*H-Ind), 113.47 (1C, *C*-Ind), 116.62 (1C, C-Imi), 117.68 (1C, *C*H-Ind), 120.62 (1C, *C*H-Ind), 122.81 (1C, *C*H-Ind), 123.41 (1C, *C*HN-Ind), 125.43 (1C, *C*-Ind); 128.29 (1C, C-5), 128.52 (1C, C-6), 129.20 (1C, C-8), 130.65 (1C, C-8a), 131.76 (1C, C-7), 132.12 (1C, C-4a), 132.33 (2C, *C*H-Imi), 136.33 (1C, *C*-Ind), 163.44 (1C, C-1), 167.09 (1C, *C*ON). ESI-HRMS (m/z) calculated for [M + H]^+^ ion species C_24_H_23_N_4_O_3_: 415.1765; found 415.1762.

##### *rel*-(3*R*,4*R*)-3-(1*H*-indol-3-yl)-2-(2-methoxyethyl)-4-(4-methylpiperazine-1-carbonyl)-3,4-dihydroisoquinolin-1(2*H*)-one (**4e**)

In this case, *N*-methylpiperazine (0.11 mL, 1 mmol) was used as an amine. After the completion of the reaction (1 h), the residual oil was triturated with ethyl acetate 3 mL and the crystals formed were collected by filtration. Yield 0.160g (72%) of off-white crystals, m.p. 233–235 °C.

^1^H NMR (CDCl_3_, 400,23 MHz) δ 1.25–1.55 (m, 1H, C*H*_2_-pip), 1.87–2.05 (m, 1H, C*H*_2_-pip), 2.17 (s, 3H, NC*H*_3_), 2.22–2.36 (m, 1H, C*H*_2_-pip), 2.42–2.60 (m, 1H, C*H*_2_-pip), 3.10 (ddd, 1H, NC*H*_2_, *J* = 5.1, 7.7, 14.0 Hz), 3.18 (s, 3H, OC*H*_3_), 3.40 (ddd, 1H, C*H*_2_O, *J* = 4.8, 5.1, 9.8 Hz), 3.43–3.69 (m, 4H, 1xC*H*_2_O, 3xC*H*_2_-pip), 3.73–3.96 (m, 1H, C*H*_2_-pip), 4.21 (ddd, 1H, NC*H*_2_, *J* = 4.8, 4.8, 14.0 Hz), 4.69 (d, 1H, H-4, *J* = 7.4 Hz), 5.45 (d, 1H, H-3, *J* = 7.4 Hz), 6.83–6.90 (m, 2H, H-5, C*H*N-Ind), 7.07 (ddd, 1H, C*H*-Ind, *J* = 0.7, 6.9, 7.9 Hz), 7.15 (ddd, 1H, C*H*-Ind, *J* = 0.7, 6.9, 8.0 Hz), 7.31–7.40 (m, 3H, C*H*-Ind, H-6, H-7), 7.55 (d, 1H, C*H*-Ind, *J* = 7.9 Hz), 8.10–8.17 (m, 1H, H-8), 8.81 (br. s, 1H, N*H*); ^13^C NMR (CDCl_3_, 100.64 MHz) δ 40.98 (1C, *C*H_2_-pip), 44.24 (1C, N*C*H_2_), 44.84 (1C, N*C*H_3_), 45.04 (1C, *C*H_2_-pip), 46.56 (1C, C-4), 53.85 (1C, *C*H_2_-pip), 54.25 (1C, *C*H_2_-pip), 57.42 (1C, C-3), 59.03 (1C, O*C*H_3_), 70.99 (1C, *C*H_2_O), 112.20 (1C, *C*H-Ind), 113.08 (1C, *C*-Ind), 118.71 (1C, *C*H-Ind), 120.60 (1C, *C*H-Ind), 122.78 (1C, *C*H-Ind), 124.73 (1C, *C*HN-Ind), 125.55 (1C, *C*-Ind); 126.53 (1C, C-5), 128.13 (1C, C-6), 128.63 (1C, C-8), 129.90 (1C, C-8a), 132.33 (1C, C-7), 135.53 (1C, C-4a), 136.64 (1C, *C*-Ind), 164.68 (1C, C-1), 169.62 (1C, *C*ON). ESI-HRMS (m/z) calculated for [M + H]^+^ ion species C_26_H_31_N_4_O_3_: 447.2396; found 447.2488.

#### 3.2.4. rel-(3*R*,4*R*)-2-hexyl-1-oxo-3-(pyridin-2-yl)-1,2,3,4-tetrahydroisoquinoline-4-carboxylic acids and *rel*-(3*S*,4*R*)-2-hexyl-1-oxo-3-(pyridin-2-yl)-1,2,3,4-tetrahydroisoquinoline-4-carboxylic acids (*trans*-**6** and *cis*-**6**)

A solution of imine **5** (2.5 g, 0.013 mol) in dry 1,2-dichloroethane (8 mL) was added dropwise to a suspension of homophthalic anhydride (**1**, 2.12 g, 0.013 mol) in dry 1,2-dichloroethane (15 mL). The reaction mixture was stirred 1 h at room temperature and the consumption of the anhydride **1** was established by TLC. Then the reaction mixture was diluted with dichloromethane and washed three times with 10% sodium carbonate. The alkaline solution was acidified with diluted hydrochloric acid (1:1) to pH 3–4 and extracted three times with ethyl acetate. The combined organic layers were washed with water, dried with anhydrous sodium sulfate, and the solvent was evaporated under reduced pressure. The crude product was obtained in 87% yield and the acids *trans*-**6** and *cis*-**6** were separated by column chromatography (petroleum ether: ethyl acetate: formic acid 3.5: 1.5: 0.06). 

The yield of *cis*-**6** could not be determined because of the observed rapid epimerization to *trans*-**6** in solution.

^1^H NMR (DMSO-d_6_, 250.13 MHz) δ: signals for *cis*-**6**: δ 2.81–2.84 (m, 1H, H^a^-NC*H*_2_), 3.70–3.72 (m, 1H, H^b^-NC*H*_2_), 4.73 (d, 1H, *H*-4, *J* = 6.1 Hz), 5.19 (d, 1H, *H*-3, *J* = 6.1 Hz), 7.03 (d, 1H, C*H*-Ph, *J* = 7.9 Hz), 7.98 (d, 1H, H-8, *J* = 7.2 Hz), 8.34 (d, 1H, C*H*-Pyr, *J* = 4 Hz); signals for *trans*-**6**: δ 2.73–2.79 (m, 1H, H^a^-NC*H*_2_), 4.03–4.12 (m, 1H, H^b^-NC*H*_2_), 4.41 (d, 1H, H-4, *J* = 1.0 Hz), 5.33 (d, 1H, H-3, *J* = 1.0 Hz), 7.01 (d, 1H, C*H*-Ph, *J* = 7.9 Hz), 7.87 (dd, 1H, H-8, *J* = 3.0, 6.0 Hz), 7.98 (dm, 1H, C*H*-Pyr, *J* = 4.8 Hz); signals for the both of diastereomers: δ 0.80–0.87 (m, 3H, C*H*_3_), 1.21–1.24 (m, 6H, 3xC*H*_2_), 1.40–1.53 (m, 2H, C*H*_2_), 7.17–7.23 (m, 2H, C*H*-*Ph*, C*H*-*Pyr*), 7.30–7.46 (m, 2H, C*H-Pyr*), 7.62–7.69 (m, 1H, C*H*-*Ph*).

#### 3.2.5. *rel*-(3*R*,4*R*)-methyl 2-hexyl-1-oxo-3-(pyridin-2-yl)-1,2,3,4-tetrahydroisoquinoline-4-carboxylate (*trans*-**7**) 

Compound *trans*-**7** was obtained following the procedure described in the literature [28].

#### 3.2.6. *rel*-(3*R*,4*R*)-2-hexyl-4-(hydroxymethyl)-3-(pyridin-2-yl)-3,4-dihydroisoquinolin-1(2*H*)-one (*trans*-**8**)

A solution of the ester *trans*-**7** (2 g, 0.005 mol) in dry tetrahydrofurane (6 mL) was added dropwise (15–20 min) to a suspension of KBH_4_ (0.737 g, 0.014 mol) and LiCl (0.579 g, 0.014 mol) in tetrahydrofurane (4 mL). The reaction mixture was stirred at room temperature for 20 h or was sonicated for 1.5 h. The consumption of the ester *trans*-**7** was established by TLC. The reaction mixture was concentrated under reduced pressure, poured into water, and extracted three times with ethyl acetate. The combined organic layers were washed with water, dried with anhydrous sodium sulfate, and the solvent was evaporated under reduced pressure. The crude product was obtained in 88% yield as a yellow oil, which crystallized in ethyl acetate. Yield = 1.35 g (73%, the yield is the same under the both reaction conditions investigated); m.p. 110–112 ^o^C (white crystals, recryst. from ethyl acetate).

^1^H NMR (CDCl_3_, 500,13 MHz) δ 0.86 (t, 3H, C*H*_3_, *J* = 7.1 Hz), 1.27–1.38 (m, 6H, 3xC*H*_2_), 1.66–1.73 (m, 2H, C*H*_2_), 2.00 (brs, 1H, O*H*), 2.78 (ddd, 1H, NC*H*_2_, *J* = 5.7, 8.9, 13.4 Hz), 3.53 (ddd, 1H, H-3, *J* = 1.1, 6.0, 9.3 Hz), 3.81–3.88 (m, 2H, OC*H*_2_), 4.29 (ddd, 1H, NC*H*_2_, *J* = 7.0, 9.0, 13.4 Hz), 5.16 (s, 1H, H-3), 6.97 (d, 1H, C*H*-*Ph*, *J* = 7.8 Hz), 7.03–7.04 (m, 1H, C*H*-*Pyr*), 7.10 (ddd, 1H, C*H*-*Pyr*, *J* = 0.8, 4.8, 7.5 Hz), 7.33 7.50 (dt, 1H, C*H*-*Ph*, *J* = 1.8, 7.7 Hz), 8.15 (t, 1H, H-8, *J* = 6.5 Hz), 8.53 (ddd, 1H, C*H*-*Pyr*, *J* = 1.0, 1.7, 4.8 Hz).^13^C NMR (CDCl_3_, 62.90 MHz) δ 14.03 (1C-*C*H_3_), 22.59 (1C-*C*H_2_), 26.76 (1C-*C*H_2_), 27.95 (1C-*C*H_2_), 31.59 (1C-*C*H_2_), 46.98 (1C-*C*H), 47.23 (1C-*C*H_2_N), 61.37 (1C-*C*H), 65.13 (1C-*C*H_2_O), 127.82 (1C-*C*H), 128.13 (1C-*C*H), 128.30 (3C-*C*H), 129.01 (1C-*C*), 131.96 (1C-*C*H), 135.65 (1C-*C*), 136.95 (1C-*C*H), 149.64 (1C-*C*H), 159.91 (1C-*C*), 163.97 (1C-*C*ON). Calculated (%) for C_21_H_26_N_2_O_2_ (338.45): C 74.53, H 7.74. Found (%): C 74.59, H 7.91.

#### 3.2.7. *rel*-2-(((3*S*,4*R*)-2-hexyl-1-oxo-3-(pyridin-2-yl)-1,2,3,4-tetrahydroisoquinolin-4-yl)methyl)isoindoline-1,3-dione (*trans*-**9**)

Diethylazodicarboxylate (DEAD) solution 40 wt.% in toluene (5.6 mL, 0.012 mol) was added dropwise to a stirred suspension of *trans*-**8** (3.18 g, 0.009 mol), triphenyl phosphine (3.20 g, 0.012 mol), and phthalimide (1.79 g, 0.012 mol) in dry tetrahydrofurane (20 mL) at 5–10 ^o^C and in an inert atmosphere. After the addition of DEAD, the reaction mixture was stirred at room temperature for 20 h. The consumption of the alcohol *trans*-**8** was established by TLC. The solvents were evaporated under reduced pressure and the residue was purified by column chromatography (dichloromethane: ethyl acetate 4.6:0.4). Compound *trans*-**9** was obtained as an oil. Yield = 4.17 g (95%).

^1^H NMR (CDCl_3_, 600.18 MHz) δ 0.78 (t, 3H, *C*H_3_, *J* = 7.0 Hz), 1.20–1.33 (m, 6H, 3xC*H*_2_), 1.57–1.72 (m, 2H, C*H*_2_), 3.15 (ddd, 1H, NC*H*_2_, *J* = 5.3, 10.3, 13.5 Hz), 3.70 (ddd, 1H, NC*H_2_*, *J* = 0.9, 5.8, 8.2 Hz), 3.78 (dd, 1H, H-4, *J* = 5.8, 13.7 Hz), 3.96 (ddd, 1H, NC*H*_2_, *J* = 6.1, 10.1, 13.5 Hz), 4.12–4.15 (m, 1H, NC*H*_2_), 4.75 (s, 1H, H-3), 6.82 (d, 1H, C*H*-*Ph*, *J* = 7.9 Hz), 6.94 (d, 1H, C*H*-*Pyr*, *J* = 7.2 Hz), 6.98 (dd, 1H, C*H*-*Pyr*, *J* = 4.9, 6.8 Hz), 7.20 (dt, 1H, C*H*-*Ph*, *J* = 1.5, 7.5 Hz), 7.26 (dt, 1H, C*H*-*Ph*, *J* = 1.1, 7.5 Hz), 7.36 (dt, 1H, C*H*-*Pyr*, *J* = 1.8, 7.8 Hz), 7.65–7.67 (m, 2H, C*H*-*Phth*), 7.78–7.81 (m, 2H, CH-*Phth*), 8.10 (d, 1H, H-8, *J* = 1.3 Hz), 8.38 (d, 1H, C*H*-*Pyr*, *J* = 0.8, 4.8 Hz). ^13^C NMR (CDCl_3_, 150 MHz) δ 14.05 (1C-*C*H_3_), 22.58 (1C-*C*H_2_), 26.90 (1C-*C*H_2_), 27.82 (1C-*C*H_2_), 31.55 (1C-*C*H_2_), 42.19 (1C-*C*H), 43.60 (1C-*C*H_2_N), 48.15 (1C-*C*H_2_N), 64.20 (1C-*C*H), 120.17 (1C-*C*H), 122.34 (1C-*C*H), 123.50 (3C-*C*H), 128.06 (1C-*C*H), 128.09 (1C-*C*H), 128.19 (1C-*C*H), 128.93 (1C-*C*), 131.97 (1C-*C*H), 132.16 (1C-*C*H), 134.15 (2C-*C*H), 135.87 (1C-*C*), 136.64 (1C-*C*), 149.56 (1C-*C*), 159.44 (1C-*C*), 163.90 (1C-*C*ON), 168.19 (1C-*C*ON). Calculated (%) for C_29_H_29_N_3_O_3_ (467.57): N 8.99. Found (%): N 8.79.

#### 3.2.8. *rel*-(3*R*,4*S*)-4-(aminomethyl)-2-hexyl-3-(pyridin-2-yl)-3,4-dihydroisoquinolin-1(2*H*)-one (*trans*-**10**)

Compound *trans*-**9** (4.17 g, 0.009 mol) was heated at 80–90 °C for 2 h with ethylenediamine (7.5 mL, 0.112 mol). After the end of the reaction, the reaction mixture was cooled down to room temperature, poured into brine, and extracted three times with ethyl acetate. The combined organic layers were washed with water, dried with anhydrous sodium sulfate, and the solvent was evaporated under reduced pressure. The crude product was obtained as an oil and purified by column chromatography (*tert*-butyl methyl ether: propan-2-ol: ammonium hydroxide 4:1:0.06). Compound *trans*-**10** was isolated as an oil difficult to obtain in absolutely anhydrous form. Yield: 2.19 g (70%).

^1^H NMR (CDCl_3_, 250.13 MHz) δ 0.85–0.90 (m, 3H, C*H*_3_), 1.27–1.39 (m, 6H, 3xC*H*_2_), 1.62–1.80 (m, 2H, H-C*H*_2_), 2.72–2.83 (m, 1H, H^a^-NC*H*_2_), 2.99–3.04 (m, 2H, NC*H*_2_), 3.25–3.28 (m, 1H, H-4), 4.22–4.31 (m, 1H, H^b^-NC*H*_2_), 4.75 (s, 1H, H-3),6.92 (d, 1H, C*H*-*Ph*, *J* = 8 Hz), 6.99–7.12 (m, 2H, C*H*-*Ph*, C*H*-Pyr), 7.29–7.35 (m, 2H, C*H*-*Pyr*), 7.49 (dt, 1H, C*H*-*Ph*, J = 7.8 Hz), 8,14 (dd, 1H, H-8, *J* = 2.8, 5.8 Hz), 8.54 (dm, 1H, C*H*-*Pyr*, *J* = 4.8 Hz). ^13^C NMR (CDCl_3_, 62.90 MHz) δ 13.9 (1C-*C*H_3_), 22.5 (1C-*C*H_2_), 25.3 (1C-*C*H_2_), 26.6 (1C-*C*H_2_), 27.9 (1C-*C*H_2_), 31.5 (1C-*C*H), 46.8 (1C- *C*H_2_N), 47.7 (1C-*C*H_2_N), 62.8 (1C-*C*H), 119.9 (1C-*C*H), 122.1 (1C-*C*H), 127.5 (1C-*C*H), 128.0 (1C-*C*H), 128.1 (1C-*C*H), 128.5 (1C-*C*), 131.7 (1C-*C*H), 136.7 (1C-*C*H), 137.2 (1C-*C*), 146.9 (1C-*C*H), 159.8 (1C-*C*), 163.9 (1C-*C*ON).

#### 3.2.9. (*S*)-*N*-((3*R*,4*S*)-2-hexyl-1-oxo-3-(pyridin-2-yl)-1,2,3,4-tetrahydroisoquinolin-4-yl)methyl)-3-phenyl-2-(2,2,2-trifluoroacetamido)propanamide and (*S*)-*N*-((3*S*,4*R*)-2-hexyl-1-oxo-3-(pyridin-2-yl)-1,2,3,4-tetrahydroisoquinolin-4-yl)methyl)-3-phenyl-2-(2,2,2-trifluoroacetamido)propanamide (*trans*-**11a** + *trans*-**11b**)

To the suspension of *N*-trifluoroacetyl-l-phenylalanine (Tfa-Phe-OH) 0.131 g (0.5 mmol) in benzene, thionyl chloride 0.1 mL (1.4 mmol) was added dropwise under stirring. The reaction mixture was stirred at 90 °C for 2 h. Then the excess of benzene and thionyl chloride were distilled off under vacuum. A solution of compound *trans*-**10** (0.29 g, 0.8 mmol) in tetrahydrofurane (1.5 mL) was added dropwise to the resulting solid of *N*-trifluoroacetyl-l-phenylalanine chloride at 5–10 °C. The reaction mixture was stirred for 30 min. Then the reaction mixture was diluted with ethyl acetate and washed several times with water to neutral pH. The organic layer was dried with anhydrous sodium sulfate and the solvent was evaporated under reduced pressure. The resulting crude product was obtained as an oil, purified by column chromatography (petroleum ether: ethyl acetate 1:1) and subsequent recrystallisation (petroleum ether: ethyl acetate 4:1). Yield: 0.24 g (80%).

^1^H NMR (CDCl_3_, 600.18 MHz) δ 0.84–0.89 (m, 6H, C*H*_3_), 1.20–1.37 (m, 12H, 3xC*H*_2_), 1.45–1.61 (m, 4H, C*H*_2_), 2.78–2.88 (m, 2H, NC*H*_2_), 3.07–3.22 (m, 5H, 1xNHC*H*_2_a, 4xCH_2_Ph), 3.32–3.38 (m, 1H, 1xNHC*H*_2_a), 3.39–3.43 (m, 1H, H-4a), 3.49–3.53 (m, 1H, H-4b), 3.71–3.77 (m, 2H, 1xNHC*H*_2_b), 3.78–3.86 (m, 2H, 1xNHC*H*_2_b), 4.05–4.17 (m, 2H, NC*H*_2_), 4.61–4.67 (m, 2H, C(O)C*H*) 4.72 (s, 2H, H-3), 6.28 (brs, 1H, N*H*), 6.32 (brs, 1H, N*H*), 6.64 (d, 1H, C*H*-*Ar*, *J* = 7.5 Hz), 6.85–6.93 (m, 3H, C*H*-*Ar*), 7.10–7.15 (m, 2H, C*H*-*Ar*), 7.21–7.35 (m, 14H, 2xC*H*-*Ar*; 10x C*H*-*Ph*); 7.39–7.45 (m, 2H, N*H*), 7.46–7.52 (m, 2H, C*H*-*Ar*); 8.08–8.13 (m, 2H, H-8), 8.49–8.52 (m, 2H, C*H*-*Ar*). ^13^C NMR (CDCl_3_, 150.92 MHz) δ 14.11 (1Ca-*C*H_3_), 14.14 (1Cb-*C*H_3_), 22.70 (2C-*C*H_2_), 26.87 (2C-*C*H_2_), 28.13 (2C-*C*H_2_), 31.65 (2C-*C*H_2_), 38.54 (1Ca-*C*H_2_), 38.77 (1Cb-*C*H_2_), 43.60 (1Ca-*C*4), 43.74 (1Cb-*C*4), 44.17 (1Ca-NH*C*H_2_), 44.26 (1Cb-NH*C*H_2_), 47.39 (1Ca-N*C*H_2_), 47.44 (1Cb-N*C*H_2_), 55.17 (1Ca-C(O)*C*H), 55.25 (1Cb-C(O)*C*H), 62.98 (1Ca-*C*3), 63.59 (1Cb-*C*3), 115.40 (q,1Ca-CF_3_, *J* = 287.6), 115.78 (q,1Cb-CF_3_, *J* = 287.6), 120.31 (1Ca-*C*H), 120.41 (1Cb-*C*H), 122.75 (2C-*C*H), 127.65 (2C-*C*H), 128.18 (1Ca-*C*H), 128.26 (2C-*C*H), 128.28 (1Cb-*C*H), 128.39 (2C-*C*H), 128.71 (1Ca-*C*H), 128.73 (1Cb-*C*H), 128.99 (2Ca-*C*H), 129.05 (2Cb-*C*H), 129.48 (2C-*C*H), 129.50 (2C-*C*H), 132.34 (1Ca-*C*H), 132.41 (1Cb-*C*H), 135.61 (1C-*C*H), 135.76 (1C-*C*H), 135.91 (1Ca-*C*H), 135.94 (1Cb-*C*H), 137.29 (1Ca-*C*), 137.42 (1Cb-*C*), 149.42 (1Ca-*C*), 149.50 (1Cb-*C*), 156.87 (q,1Ca-COCF_3_, *J* = 38.1 Hz), 156.99 (q,1Cb-COCF_3_, *J* = 37.9 Hz), 159.00 (1Ca-*C*ON), 159.03 (1Cb-*C*ON), 163.93 (1Ca-*C*ON), 163.95 (1Cb-*C*ON), 169.57 (1Ca-*C*ON), 169.63 (1Ca-*C*ON). Calculated (%) for C_32_H_35_F_3_N_4_O_3_ (580.66): C 66.19, H 6.08. Found (%): C 66.23, H 6.34.

#### 3.2.10. Acylation of *trans*-**10** through the Carbodiimide Method (General Procedure)

Compound *trans*-**10** 0.2 g (0.6 mmol) and the corresponding amino acid (0.6 mmol) were mixed in dry dichloroethane (2 mL). *N*,*N*’-Dicyclohexylcarbodiimide (DCC, 0.161 g, 0.8 mmol) was added in portions to the stirred reaction mixture and under cooling (−10 °C). The reaction mixture was stirred at -2 to 0 °C for 2 h. Then the resulting precipitate obtained at the 10th minute from the start of the reaction was filtered, and the filtrate was concentrated under reduced pressure. The resulting oily residue was dissolved in ethyl acetate and washed successively with hydrochloric acid (1:1), water, Na_2_CO_3_ (10% aqueous solution), and again with water. The organic layer was dried with anhydrous sodium sulfate and the solvent was evaporated under reduced pressure. The resulting crude product was obtained as an oil, purified by column chromatography and subsequent recrystallisation in some of the cases reported.

##### (*S*)-*N*-(((3*R*,4*S*)-2-hexyl-1-oxo-3-(pyridin-2-yl)-1,2,3,4-tetrahydroisoquinolin-4-yl)methyl)-3-phenyl-2-(2,2,2-trifluoroacetamido)propanamide and (*S*)-*N*-(((3*S*,4*R*)-2-hexyl-1-oxo-3-(pyridin-2-yl)-1,2,3,4-tetrahydroisoquinolin-4-yl)methyl)-3-phenyl-2-(2,2,2-trifluoroacetamido)propanamide (*trans*-**11a** + *trans*-**11b**)

We obtained, from *trans*-**10** (0.2 g, 0.6 mmol), *N*-trifluoroacetyl-l-phenylalanine (Tfa-Phe-OH, 0.157 g, 0.6 mmol) and DCC (0.161 g, 0.8 mmol). The resulting crude product was purified by column chromatography (petroleum ether: ethyl acetate 1:1) and subsequent recrystallisation (petroleum ether: ethyl acetate 4:1). Yield: 0.23 g (78%).

##### *tert*-butyl (*S*)-2-(((3*R*,4*S*)-2-hexyl-1-oxo-3-(pyridin-2-yl)-1,2,3,4-tetrahydroisoquinolin-4-yl)methyl)carbamoyl)pyrrolidine-1-carboxylate and *tert*-butyl (*S*)-2-(((3*S*,4*R*)-2-hexyl-1-oxo-3-(pyridin-2-yl)-1,2,3,4-tetrahydroisoquinolin-4-yl)methyl)carbamoyl)pyrrolidine-1-carboxylate (*trans*-**12a** + *trans*-**12b**)

We obtained, from *trans*-**10** (0.2 g, 0.6 mmol), *N*-BOC-l-proline (Boc-Pro-OH), 0.129 g, 0.6 mmol), and DCC (0.161 g, 0.8 mmol). The resulting crude product was purified by column chromatography (petroleum ether: ethyl acetate 2.2:2.8). Yield: 0.225 g (70%).

^1^H NMR (CDCl_3_, 600.18 MHz) δ 0.84–0.89 (m, 6H, C*H*_3_), 1.20–1.37 (m, 16H, 3xC*H*_2_, 1xC*H*_2_-*Pyrrolidine*), 1.41 (s, 9H, C(C*H*_3_)_3_a), 1.42 (s, 9H, C(C*H*_3_)_3_b), 1.58–1.69 (m, 4H, C*H*_2_), 1.83–1,98 (m, 4H, C*H*_2_-*Pyrrolidine*), 2.79–2.93 (m, 2H, NC*H*_2_), 3.22–3.50 (m, 6H, 2xC*H*_2_-*Pyrrolidine*, 2xNHC*H*_2_), 3.51–3.55 (m, 1H, H-4a), 3.56–3.60 (m, 1H, H-4b), 3.63–3.77 (m, 2H, NHC*H*_2_), 4.13–4.22 (m, 2H, NC*H*_2_), 4.23–4.40 (m, 2H, C*H*-*Pyrrolidine*), 4.80 (br.s., 2H, H-3), 6.89–6.93 (m, 2H, C*H*-*Ar*), 6.94–7.03 (m, 2H, C*H*-*Ar*), 7.04–7.09 (m, 2H, C*H*-*Ar*), 7.23 (brs, 1H, N*H*a), 7.28–7.38 (m, 5H, 4xC*H*-*Ar*, N*H*b), 7.42–7.48 (m, 2H, C*H*-*Ar*), 8.09–8.15 (m, 2H, H-8), 8.45–8.51 (m, 2H, CH-*Pyr*). ^13^C NMR (CDCl_3_, 150.92 MHz) δ 14.15 (2C-*C*H_3_), 22.70 (1C-*C*H_2_a), 22.72 (1C-*C*H_2_b), 24.73 (2C-CH_2_), 26.89 (1Ca-*C*H_2_), 26.93 (1Cb-*C*H_2_), 28.12 (2C-*C*H_2_), 28.52 (6C-*C*H_3_), 29.81 (2C-*C*H_2_), 31.67 (2C-*C*H_2_), 43.81 (1C-*C*4a), 43.98 (1C-*C*4b), 44.16 (2C-*C*H_2_), 47.26 (2C-N*C*H_2_), 47.34 (2C-*C*H_2_-*Pyrrolidine*), 60.19 (2C-*C*H), 63.59 (2C-*C*3), 80.50 (1C-*C*(CH_3_)_3_), 80.67 (1C-*C*(CH_3_)_3_), 120.21 (2C-*C*H), 122.44 (2C-*C*H), 127.91 (2C-*C*H), 128.24 (6C-*C*H), 128.84 (1C-*C*a), 128.88 (1Cb-*C*), 132.01 (1C-*C*Ha), 132.12(1C-*C*Hb), 136.87 (2C-*C*H), 149.58 (1C-*C*Ha), 149.62, 156.12 (2C-*C*), (1C-*C*Hb), 159.45 (2C-*C*ON), 164.08 (2C-*C*ON), 172.57 (2C-*C*ON). Calculated (%) for C_31_H_42_N_4_O_4_ (534.32): C 69.64, H 7.92. Found (%): C 69.35, H 8.22.

##### (*S*)-*N*-(((3*R*,4*S*)-2-hexyl-1-oxo-3-(pyridin-2-yl)-1,2,3,4-tetrahydroisoquinolin-4-yl)methyl)-4-(methylthio)-2-(2,2,2-trifluoroacetamido)butanamide and (*S*)-*N*-(((3*S*,4*R*)-2-hexyl-1-oxo-3-(pyridin-2-yl)-1,2,3,4-tetrahydroisoquinolin-4-yl)methyl)-4-(methylthio)-2-(2,2,2-trifluoroacetamido)butanamide (*trans*-**13a** + *trans*-**13b**)

We obtained, from *trans*-**10** (0.2 g, 0.6 mmol), *N*-trifluoroacetyl-l-methionine (Boc-Pro-OH), 0.147 g, 0.6 mmol) and DCC (0.161 g, 0.8 mmol). The resulting crude product was purified by column chromatography (petroleum ether: ethyl acetate 1:1) and subsequent recrystallisation (petroleum ether: ethyl acetate 4:1). Yield: 0.25 g (74%).

^1^H NMR (CDCl_3_, 600.18 MHz) δ 0.83–0.89 (m, 6H, C*H*_3_), 1.20–1.40 (m, 12H, 3xC*H*_2_), 1.55–1.66 (m, 4H, C*H*_2_), 2.05–2.22 (m, 10H, 2xSC*H*_3_, 2xSC*H*_2_), 2.53–2.67 (m, 4H, 2xC*H*_2_), 2.82–2.92 (m, 2H, NC*H*_2_), 3.35–3.42 (m, 1H, NHC*H*_2_a), 3.44–3.50 (m, 1H, NHC*H*_2_b), 3.53–3.63 (m, 2H, *H*4), 3.77–3.83 (m, 1H, NHC*H*_2_b ), 3.83–3.91 (m, 1H, NHC*H*_2_a), 4.11–4.21 (m, 2H, NC*H*_2_), 4.64–4.71 (m, 2H, C(O)C*H*), 4.82 (s, 1H, H-3a), 4.83 (s, 1H, H-3b), 6.86 (brs, 1H, N*H*a), 6.89–6.94 (m, 2H, C*H*-*Ar*), 7.96–7.03 (m, 3H, 2xC*H*-*Ar,* N*H*b), 7.10–7.16 (m, 2H, C*H*-*Ar*), 7.33–7.39 (m, 4H, C*H*-*Ar*), 7.47–7.55 (m, 3H, 2xC*H*-Ar, N*H*a), 7.59 (d, 1H, N*H*b, *J* = 7.5 Hz), 8.10–8.15 (d, 2H, H-8), 8.47–8.54 (m, 2H, C*H*-*Pyr*). ^13^C NMR (CDCl_3_, 150.92 MHz) δ 14.12 (1C-*C*H_3_a), 14.14 (1C-*C*H_3_b), 15.17 (1C-S*C*H_3_a), 15.28 (1C-S*C*H_3_b), 22.70 (2C-*C*H_2_), 26.89 (2C-*C*H_2_), 28.18 (2C-*C*H_2_), 29.84 (2C-*C*H_2_), 31.36 (1C-S*C*H_2_a), 31.42 (1C-S*C*H_2_b), 31.66 (2C-*C*H_2_), 43.75 (1C-*C*4a), 43.80 (1C-*C*4b), 44.30 (1C-NH*C*H_2_a), 44.41 (1C-NH*C*H_2_b), 47.37 (1C-N*C*H_2_a), 47.43 (1C-N*C*H_2_b), 52.67 (1C-*C*Ha), 52.75 (1C-*C*Hb), 63.16 (1C-*C*3a), 63.52 (1C-*C*3b), 115.81 (q,2C-*C*F_3_, *J* = 287.9), 120.35 (1C-*C*Ha), 120.35 (1C-*C*Hb), 122.81 (2C-*C*H), 128.35 (3C-*C*H), 128.38 (1C-*C*H), 128.42 (1C-*C*H), 128.43 (1C-*C*H), 128.70 (1C-*C*), 128.82 (1C-*C*), 132.43 (1C-*C*Ha), 132.48 (1C-*C*Hb), 135.88 (2C-*C*), 137.41 (1C-*C*Ha), 137.56 (1C-*C*Hb), 149.50 (2C-*C*H), 157.07 (q,1Ca-COCF_3_, *J* = 37.5 Hz), 157.25 (q,1Cb-COCF_3_, *J* = 37.7 Hz), 159.01 (2C-C), 163.93 (1C-*C*ONa), 164.00 (1C-*C*ONb), 169.77 (1C-*C*ONa), 169.79 (1C-*C*ONb). Calculated (%) for C_28_H_35_F_3_N_4_O_3_S (564.66): C 59.56, H 6.25. Found (%): C 59.35, H 6.52.

### 3.3. Microbiology

#### 3.3.1. Cytotoxicity Assay 

Confluent monolayer cell culture in 96-well plates (Costar®, Corning Inc., Kennebunk, ME, USA) was treated with 0.1 mL/well containing a maintenance medium that did not contain (or contained decreasing) concentrations of test substances. The cells were incubated at 37 °C and 5% CO_2_ for 5 days. After microscopic evaluation, the medium containing the test compound was removed, the cells were washed and incubated with neutral red at 37 °C for 3 h. After incubation, the neutral red dye was removed and the cells were washed with PBS, and 0.15 mL/well desorbing solution (1% glacial acetic acid and 49% ethanol in distilled water) was added. The optical density (OD) of each well was read at 540 nm in a microplate reader (Biotek Organon, West Chester, PA, USA). Fifty percent cytotoxic concentration (CC_50_) was defined as the concentration of the material that reduces cell viability by fifty percent compared to untreated controls. Each sample was tested in triplicate with four wells for cell culture on a test sample.

The maximum tolerable concentration (MTC) of the extracts was also determined, which is the concentration at which they do not affect the cell monolayer, and, in the sample, it looks like the cells in the control sample (untreated with compounds).

#### 3.3.2. Antiviral Activity Assay

The cytopathic effect (CPE) inhibition test was used for assessment of antiviral activity of the tested compounds [33]. Confluent cell monolayer in 96-well plates was infected with 100 cell culture infectious dose 50% (CCID_50_) in 0.1 mL (coronavirus OC-43 or 229E strain). After 120 min of virus adsorption, the tested compound was added in various concentrations and cells were incubated for 5 day at 33 °C for OC-43 and at 35 °C for 229E strain. The cytopathic effect was determined using a neutral red uptake assay, and the percentage of CPE inhibition for each concentration of the test sample was calculated using the following formula: % CPE = [OD_test sample_ − OD_virus control_]/[OD_toxicity control_ − OD_virus control_] × 100
where OD_test sample_ is the mean value of the ODs of the wells inoculated with virus and treated with the test sample in the respective concentration, OD_virus control_ is the mean value of the ODs of the virus control wells (with no compound in the medium) and OD_toxicity control_ is the mean value of the ODs of the wells not inoculated with virus but treated with the corresponding concentration of the test compound. The 50% inhibitory concentration (IC_50_) was defined as the concentration of the test substance that inhibited 50% of viral replication when compared to the virus control. The selectivity index (SI) was calculated from the ratio CC_50_/IC_50._

#### 3.3.3. Pre-Treatment of Healthy Cells

Cell monolayers (HCT-8 or MRC-5 cell culture) grown in 24-well cell culture plates (CELLSTAR, Greiner Bio-One) were treated for different time intervals, 15, 30, 60, 90, and 120 min, at maximum tolerable concentration (MTC) of heterocyclic compounds in maintenance medium (1 mL/well). After the above time intervals, the compounds were removed and the cells were washed with phosphate buffered saline (PBS) and inoculated with human coronavirus strain OC-43 or 229E (1000 CCID_50_ in 1 mL/well). After 120 min of adsorption, the non-adsorbed virus was removed and the cells were coated with a maintenance medium. Samples were incubated at 33 °C for OC-43 and 35 °C for the 229E strain for 120 h followed by three times freezing and thawing; infectious virus titers were determined by the final dilution method. Δlg was determined compared to the viral control (untreated with the compounds). Each sample was prepared in four replicates. Chloroquine and hydroxychloroquine were used as reference substances.

#### 3.3.4. Effect on Viral Adsorption

24-well plates containing monolayer cell culture from HCT-8/MRC-5 cells were pre-cooled to 4 °C and inoculated with 100 CCID_50_ of human coronavirus (OC-43/229E). In parallel, they were treated with heterocyclic compounds at their maximum tolerance concentration (MTC) and incubated at 4 °C for the time of virus adsorption. Chloroquine was used as the reference substance. At various time intervals (15, 30, 60, 90, and 120 min), the cells were washed with PBS to remove both the compound and the unattached virus, then coated with support medium and incubated at 33 °C for OC-43 and 35 °C for the 229E strain for 120 h. After three times freezing and thawing, the infectious viral titer of each sample was determined by the final dilution method. Δlg was determined compared to the viral control (untreated with the compounds). Each sample was prepared in four replicates.

## 4. Conclusions

Novel 1-oxo-1,2,3,4-tetrahydroisoquinoline derivatives containing an amide or ami-domethyl function in the 4-th isoquinoline core position were synthesized through two different synthetic pathways. Selected representatives alongside previously synthesized 4-aminomethyltetrahydroisoquinolin-1-ones, piperidinone, and thiomorpholinone were included in a comparative preliminary analysis of their antiviral activity against the replication of two strains of human coronavirus: 229E and OC-43. 

The antiviral effect of some of the heterocyclic compounds, when administered after viral adsorption and the virus had already penetrated the host cell, indicated that these substances specifically affect stages of the intracellular replicative cycle of coronavirus strains. It was shown by QSAR analysis that the electrostatic interactions between the respective molecule and the target biomolecule play an important role in the mechanism of action. Some of the compounds tested showed activity close to, or even higher than, that of chloroquine. 

Our experiments show that compounds with a similar structure could be used to reduce viral yield in coronavirus infection and inhibit the stage of attachment of the virus to the host cell, as well as to protect healthy cells. Moreover, the conducted biological studies directed us to a leading THIQ-based structure, **Avir**-**7**; the appropriate future modifications of which would allow the preparation of its analogues with higher anti-coronavirus potential.

## Data Availability

Not applicable.

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
