# Peer review of "Synthesis of Novel 1-Oxo-2,3,4-trisubstituted Tetrahydroisoquinoline Derivatives, Bearing Other Heterocyclic Moieties and Comparative Preliminary Study of Anti-Coronavirus Activity of Selected Compounds"

_molecules, 2023, doi:10.3390/molecules28031495_

Round 1

Reviewer 1 Report

The manuscript entitled “Synthesis of Novel 1-Oxo-2,3,4-trisubstituted Tetrahydroisoquinoline Derivatives, Bearing Other Heterocyclic Moieties and Comparative Preliminary Study of Anti-coronavirus Activity of Selected Compounds” by M. I. Kandinska, N. T. Burdzhiev, D. V. Cheshmedzhieva, P. P. Grozdanov, N. Vilhelmova Ilieva, N. Nikolova and I. Nikolova, was reviewed. Using well-known chemistry, the authors repeated their own work preparing some substituted tetrahydroisoquinolin-1-ones as well as some new derivatives and analogues for their evaluation as anticoronavirus agents. Since there is no novelty as regards the organic synthetic aspects of the manuscript the focus of the work is on the potential biological activity of the compounds. Although some active compounds were found, the work is very difficult to follow as there isn’t seem to be a rationale and logic in the library evaluated. The title gives the idea that a central core is maintained and “other heterocyclic moieties” are added to evaluate the structural modifications. So, for example, compounds 4 bear an indol system attached to the tetrahydroisoquinoline core, then we move on to compounds 6,7,8,9,10,11,12, and 13a/b in which a pyridine system has been included (the only different heterocycle incorporated) but many other structural aspects were also changed so comparisons will not be possible. It is also difficult to understand why all these compounds that are prepared in this study have not been included in the biological studies but compounds from previous studies were evaluated (named avir). That is to say the conclusions paragraph mentions the synthesis of new amidomethyl derivatives in the 4th position and that representative examples were evaluated but no compound from 6-13a/b was evaluated. Based on this and the issues below I consider the manuscript should be rejected.

Some other issues:

-      -    Different styles for indicating page numbers are used. For example 1669-730, then 7950-7962, then 152–61.

-         - Some refs include issue numbers others do not.

-        -  Link in ref 4 does not lead to the referenced paper.

-         - Ref 4. 3rd author should be Lion, C.

-         - Some references list their titles with all words capitalized others do not.

-       -   Ref 4. Author number 7 surname should be Andréola.

-         - Some spaces are missing between authors abbreviated names as in ref. 6.

-         - A space is missing in the title of article in ref. 4.

-          -Please check patent name ref. 9. Cannot be found.

-       -   Ref 10, different style and title missing.

-        -  Ref 13 and 14. Same journal, different abbreviation.

-        -  Refs 15 and 16 strangely lack the abbreviated second names of all authors.

-        -  Ref 16 lacks a link.

-         - Ref 19. Remove “And”

-       -   Ref 20. An author is missing.

-          -Scheme 1. Convenience.

-          -What is DCI in the manuscript main text? The scheme 1 only features DIC.

-          -Dichloroethane is abbreviated DCE not EDC as in the text.

-        -  Ref 25. Manuscript name seems to start with “Methyl..”

-       -   Line 157 gives the idea that conformation is relative configuration and this is incorrect.

-       -   Line 197 should begin with “compounds….” instead of “all compounds”.

-       -   Table 2 shows no stereochemistry.

-   -       In the text the authors compare potency of the compounds with the SI values and this leads to confusion because SI is selectivity index, values of IC should be compared instead.

-     -     2D NMR spectra should be provided for representative examples to assist structure identification.

-  -        Synthesis 3.2.1, what is acid Z mentioned?

-   -       Many spectra are not clear, they all look different with different ranges of ppm (some proton NMR start at 2 ppm) and from different sources with different styles which is odd. Some look like pasted images also. They should be clearer and feature in one page per spectrum in order to facilitate analysis.

-  -        Numbering of the compounds and addition of structures to SI would also help analysis.

-   -       Writing is poor and should be improved.

Reviewer 2 Report

The paper describes the synthesis of a series of novel tetrahydroisoquinoline derivatives for the screening of their anti-coronavirus activity. Though the synthetic part is quite trivial and, in certain parts, questionable, some peculiar results were obtained during the biotesting. The obtained results may be useful for further optimization of structure in the design of antiviral agents. From this point of view the manuscript may be published in Molecules, but the significant revision should be done.

1)     In Introduction, the examples of bioactive compounds with THIQ-scaffold should be provided. More important, previously obtained results and the goal of the present work should also be shown in scheme.

2)     Experimental details (for example, lines 65-69, 72-75) which repeat the description of the synthesis of compounds in section “Materials and Methods” should be omitted.

3)     What are the purple crystals, mentioned on line 73? Was DCI (line 75) or DIC (Scheme 1) used as a coupling agent?

4)     It is not clear how the product 4a is formed: what NMR data prove the formation of this very structure?

5)     Also unusual is the product 4b formed with the participation of methylene chloride. Are there any analogues described in literature? Which product will be formed when replacing methylene chloride with another solvent?

6)     TBTU should be probed as coupling agent (scheme 1) also for the cases of thiomorpholine, piperidine and morpholine.  All the conditions probed, including those described in lines 70-88, should be collected in a sole table, clearly describing the optimization of the conditions of coupling reaction.

7)     For all new compounds data of HRMS or elemental analysis must be given

8)     What is the structure the number trans-6a’ (Table 1) corresponds to?

9)     The result of the study of epimerization reaction mechanism is not clear. If the proposed mechanism “is not the preferred one”, more appropriate pathways should be looked for.

10)It would be useful to do primary SAR analysis, based on the found antiviral activities.

11) Scheme 3. “)))” in conditions should be corrected; yields should be given.

12) Everywhere yields should be rounded up to whole numbers: for example, 62% but not 61.6%.

13) The conditions and representation of substituents on scheme 4 should be carefully checked and corrected

14) Authors should indicate the accuracy of the measurements in table 3

15) It would be useful to introduce the entry of 0 minutes as a base standard in viral titre measurements (tables 4-7).

16)Some mistakes are present in the text (for example, line 143 “geometry…  are shown”; line 223 - “toxicity of the MRC-5 cell”, line 395 – “tiomorpholine”); therefore I would recommend to do a spell check.

Reviewer 3 Report

Kandinska and co-worker reported the Synthesis of Novel 1-Oxo-2,3,4-trisubstituted Tetrahydroiso-  quinoline Derivatives and evaluated them for their anti-viral and toxicity effects. The authors have provided lots of biological studies and depth synthetic analyses. The data provided by the authors were clean and well-presented. Overall, it’s a well-written manuscript and qualifies for publication. There are some minor comments below need to be addressed before the acceptance of this manuscript

1.      Please provide more introduction regarding the biological potential of  THIQ.

2.      Please provide the full name of coupling reagents used for synthesizing compounds e.g DIC etc.

3.      Reactions time in scheme 1 was missing.

4.      Please mention the solvent in scheme 2

5.      Why did the authors not evaluate compounds 12a-b and 13a-b for bioactivities?

6. In Table 3, replace the comma (‘) with decimal (.) for activity values.

7.      Activity values for compounds Avir-5 and 11a+11b were missing in table 3.

8.       Authors should provide the structure-activity relationship of the newly synthesized compounds which helps the reader with the future design of the molecules.

9.       Format the references according to journal guidelines.

Round 2

Reviewer 2 Report

As most of previously mentioned remarks have been addressed, I suppose, the article may be published after the following minor revisions:

1. Picture 1. For bioactive compounds, which have no trademark names, the references should be added.

2. The purple crystals that are the result of the attempt to synthesize the acid chloride of the acid 3, should be either characterized or not described at all.

3. In the answer to reviewer’s comments it has been explained, what trans-6a’ (Table 1) corresponds to, but it is still not present anywhere in the manuscript. The structure should be inserted either to main text or to supplementary materials with appropriate reference.

4. Scheme 4. Conditions vi) still do not include aminoacids. Please, correct.

Author Response

Dear reviewer,

Please find here our point-by-point response to your concerns:

Point 1: Picture 1. For bioactive compounds, which have no trademark names, the references should be added.

Answer: The clarifying references are now included in Figure 1.

Point 2: The purple crystals that are the result of the attempt to synthesize the acid chloride of the acid 3, should be either characterized or not described at all.

Answer: The purple crystals that are the result of our attempt to synthesize the acid chloride of the trans-3 are now omitted from the main text.

Point 3: In the answer to reviewer’s comments it has been explained, what trans-6a’ (Table 1) corresponds to, but it is still not present anywhere in the manuscript. The structure should be inserted either to main text or to supplementary materials with appropriate reference.

Answer: The structure of conformer trans-6a’ is now represented in Supporting Information File on Figure S40 and the corresponding reference in the main text is included.

Point 4: Scheme 4. Conditions vi) still do not include amino Please, correct.

Answer: Scheme 4 is now corrected and the amino acids used are included in conditions vi.